# Is the Treatment of the Tear Trough Deformity with Hyaluronic Acid Injections a Safe Procedure? A Systematic Review

**Salvatore D'Amato** [1] , **Romolo Fragola** [1] , **Pierfrancesco Bove** [2], **Giorgio Lo Giudice** [3] , **Paolo Gennaro** [4], **Rita Vitagliano** [1] **and Samuel Staglianò** [1,*]

1   Oral and Maxillofacial Surgery Unit, Multidisciplinary Department of Medical-Surgical and Dental Specialties, University of Campania "Luigi Vanvitelli", Via Luigi De Crecchio, 6, 80138 Naples, Italy; salvatore.damato@unicampania.it (S.D.); romolofragola@gmail.com (R.F.); rita.vitagliano@studenti.unicampania.it (R.V.)
2   Aesthetic Surgeon Private Practice, Chirurgia Della Bellezza, Via Melisurgo, 4, 80138 Naples, Italy; dr.pierfrancesco.bove@gmail.com
3   Department of Neurosciences, Reproductive and Odontostomatological Sciences, Maxillofacial Surgery Unit, University of Naples "Federico II", Via Pansini, 5, 80131 Naples, Italy; giorgio.logiudice@unina.it
4   Department of Maxillofacial Surgery, University of Siena, Strada delle Scotte, 4, 53100 Siena, Italy; paolo.gennaro@unisi.it
*   Correspondence: s.stagliano@student.unisi.it

**Abstract:** Among the various therapeutic options for the treatment of tear trough deformities, the use of hyaluronic acid-based fillers has constantly been increasing. The aim of this research is to conduct a systematic review of the published literature related to the use of hyaluronic acid-based dermal fillers for the treatment of tear trough deformities and possible related complications. A search of the published literature was conducted following the PRISMA guidelines, including PubMed, Cochrane Library, and Ovid databases. Text words and Medical Search Headings (MeSH terms) were used to identify nine articles included in our analysis. The most used filler was Restylane (Galderma). The injection technique was performed through the use of a cannula or, more frequently, with a needle, through the execution of boluses or retrograde release. The injection plane was predominantly the supra-periosteal layer. The most observed side effects were mild and included redness, edema, contour irregularities, bruising, and blue-gray dyschromia. The degree of patient satisfaction was high, with an optimal aesthetic result that was maintained for 6 to 12 months. Although the duration of treatment of tear trough deformities with HA fillers is not comparable to surgical treatment, this is a minimally invasive, safe procedure, quick to perform, and with a high degree of patient satisfaction.

**Keywords:** tear trough deformity; infraorbital hollows; soft-tissue fillers; systematic review; hyaluronic acid complication

## 1. Introduction

The tear trough, also known as the nasojugal sulcus, is the natural depression that extends inferolaterally from the medial canthus, delimited above by the infraorbital fatty bump, bounded superiorly by the infraorbital fat protuberance, whose inferior border is formed by the thick skin of the upper cheek [1–3]. Different factors can influence the aging process of the lower eyelid; for this reason, patients have a very heterogeneous clinical presentation. Age-related changes in the periorbital region include crow's feet and lower eyelid wrinkles, scleral exposure, infraorbital cavity, herniated fat pads, and excess skin of the upper and lower eyelid [1,4,5].

There are varying degrees of volume loss of the lacrimal sulcus; according to Hirmand, it is clinically possible to distinguish three classes: class I: the loss of volume is limited only medially to the lacrimal canal with or without slight flattening of the central cheek; class II: loss of both medial and lateral periorbital volume may be associated with moderate volume deficiency in the medial cheek and flattening of the upper central cheek; class III:

characterized by marked circumferential depression along the orbital rim, often associated with marked depression of the cheek and malar eminence [4].

Although our understanding of these anatomical concepts has evolved, the treatment of the lacrimal canal has still remained a challenge today. There are various techniques that can be adopted to rejuvenate this area. In the past, good results were obtained through fat grafts or through the subperiosteal placement of tear implants (Byron Medical or Implantech) [3]. However, according to the criteria described by Lambros, patients with smooth, thick skin with well-defined lacrimal sulcus can be successfully treated with HA injections [5]. Currently, the non-invasive method of HA injections is the first-choice treatment for tear deformities, all thanks to the development of new fillers that are safer, more predictable, and more affordable [6]. In general, two main classes of fillers can be distinguished: non-absorbable and resorbable ones such as HA, which can be dissolved through the use of Hyaluronidase (HYAL) [7,8].

Techniques such as these are gaining widespread acceptance, and this procedure has other desirable features: it is fast to perform and has a lasting but not permanent effect [9,10].

In fact, it is known that HA injected subcutaneously is absorbed within 1–3 years, and this is closely related to the treated area [11]; however, the bulking effects can persist in the treated area thanks to in situ neo-collagenogenesis, angiogenesis, and adipocyte proliferation in the area [12]. The onset of complications related to filler injection may be mainly due to the injection technique or the implanted material [13–15]. When HA is injected near the surface of the skin, a bluish tint, known as the "Tyndall effect", may emerge, which in persistent cases can be treated with hyaluronidase [16]. Other possible adverse effects related to HA injections are nodules, infections, granulomatous and immune-mediated reactions, edema, erythema, and ecchymosis. Of particular importance, although very rare, are the vascular complications resulting from the intravascular injection of HA [17,18].

Fortunately, vascular obstruction is a rare event and can be avoided thanks to a careful knowledge of the anatomical area to be treated and the so-called danger zones. The aim of this study is to evaluate the complications associated with treatments with resorbable HA-based fillers in general, as well as the safety of injecting HA for the treatment of tear canal deformities.

## 2. Materials and Methods

### 2.1. Eligibility Criteria

The methods and the inclusion criteria of this work were specified and documented in a protocol, according to quality standards described in the PRISMA 2020 checklist [19]. The following question was developed based on the design of the study on population, intervention, comparison, and outcome (PICO): In patients with tear trough deformities, is injection with HA a safe procedure compared to surgery?

### 2.2. Information Sources

The research was carried out on PubMed, Cochrane Library, and Ovid electronic databases identifying articles from 1 January 1957 to 2021. The search was conducted until 30 June 2021. The articles' language was limited to English using databases supplied filters.

### 2.3. Search Strategy

The keywords were used and combined with Boolean operators, adapted for every database, both as text words and Medical Search Headings (MeSH terms) as follows: HA AND complication, filler AND complication, tear trough AND filler, tear trough AND complication, hyaluronic acid AND tear trough, tear trough AND HA, tear trough AND (swelling OR bruising OR dychromia), tear troungh AND volumization, tear trough AND rejuvenation, tear trough AND non-surgical, infraorbital hollowing AND volumization, infraorbital hollowing AND rejuvenation, infraorbital hollowing AND non-surgical.

## 2.4. Study Selection

The full texts of all possibly relevant studies were selected considering the following inclusion criteria: studies in which no procedure to prevent complications was applied and English-written articles. Exclusion criteria were: articles where injection site numbers were not precisely described; articles where numbers of patients, units of hyaluronic acid applied, or type of product used were not described; and articles where the complications were not well explained. Case reports and case series with less than ten patients were excluded due to the insufficient information provided by the limited number of subjects. Review articles were excluded, but their reference lists were examined to identify other potentially pertinent studies; editorials letters and commentary were excluded. Two reviewers (R.F., S.S.) performed eligibility assessments independently. Disagreements between reviewers were resolved by consensus. When consensus was not reached, a senior member mediated (R.R.).

## 2.5. Data Collection Process

Two reviewers (G.L.G. and S.S.) performed data extraction independently. Disagreements between reviewers were resolved by consensus. When a consensus was not reached, a senior member mediated (R.R.). A standard chart form of the obtained data was prepared to facilitate comparison among the articles.

## 2.6. Data Items

The following data from each study were extracted: author, number of patients included in the study, type of hyaluronic acid filler used, injection layer, injection volume, and complications related to the procedure.

## 2.7. Study Risk of Bias Assessment

Two independent reviewers (G.L.G., S.S.) performed quality assessments of the included studies. In cases of discrepancies in the results, they consulted a third senior reviewer (R.R.). The ROBINS-I tool was used to assess non-randomized studies. Five levels (Low, Moderate, Serious, Critical, or No information) were used to present the risk of bias [20]. The Robvis visualization tool web app was used to create "traffic light" plots of the domain-level judgments for each individual result and weighted bar plots of the distribution of risk-of-bias judgments within each bias [21].

## 2.8. Summary Measures

The number of patients included in the study was expressed as integer numbers. The type of HA filler was expressed with the brand name. Injection volume was expressed in milliliter (ml). The injection layers and complications were also listed.

## 2.9. Additional Analyses

No additional analyses were performed.

## 3. Results

### 3.1. Study Selection

The PubMed search strategy identified 1684 articles for "filler AND complications", 7602 articles for "HA AND complications", 138 articles for "tear trough AND complication", 112 articles for "tear trough AND filler", 89 articles for tear "trough AND hyaluronic acid", 26 articles for "tear trough AND HA", 44 articles for "tear trough AND (swelling OR bruising OR dyschromia), 13 articles for "tear trough AND volumization", and 4 articles for "tear trough AND non-surgical". Clinical trials and randomized clinical trials were selected. The Cochrane Library search strategy identified 89 articles using "filler AND complications" as keywords, 1199 when "HA AND complications" was searched, 5 articles for "tear trough and complication", 7 articles for "tear trough and filler", 4 articles for "tear trough AND rejuvenation", 4 articles for "tear trough AND volumization", and 4

articles for "infraorbital hollowing AND volumization". Trials were filtred. The Ovid search reported no results. The total amount of articles included in the review was 11017. Trials were filtered for each database, and case reports and reviews were excluded; 4238 articles were excluded from the research because they were duplicated, and 5338 were excluded for other reasons. A total of 1441 records were screened, and 507 were excluded because they were out of topic. Nine hundred and thirty-four articles were sought for retrieval, and of these, 905 articles were screened by title, and 17 were screened for abstract. Five of them were excluded because they did not correspond to the inclusion criteria. Seven articles were considered eligible to be included in the review, and among the references of these, two studies were evaluated and defined as eligible for the study. A total of nine studies were selected as eligible at the end (Figure 1).

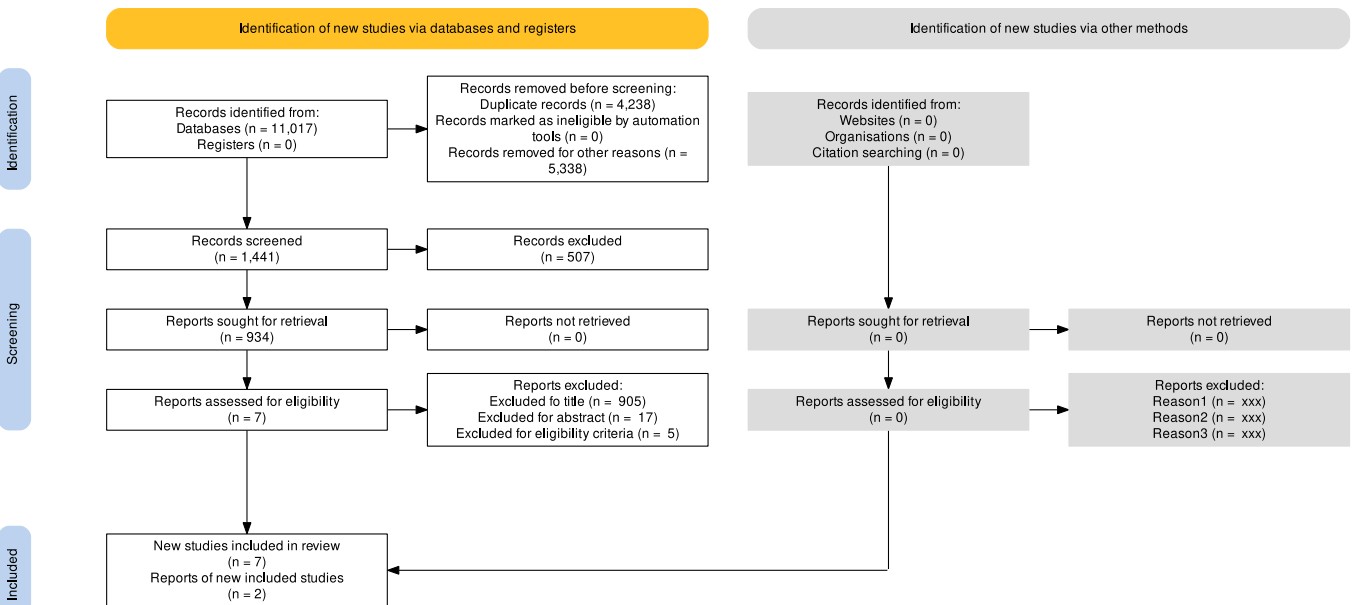

**Figure 1.** Flow diagram of the literature search and study selection.

### 3.2. Study Characteristics

A total of 830 patients treated with hyaluronic acid filler injections in the lower lid were evaluated; of these, 404 with Galderma Restylane, 175 patients were treated with Teosyal PureSense Redensity, 150 patients were treated with Juvederm Ultra Plus XC, and 101 patients were treated with Juvederm Voluma. In all studies, the outcome investigated was the enhancement of the tear trough in terms of volume, skin tone, or patient satisfaction when treated with hyaluronic acid filler. In all studies, injection layers were defined with precision; the mean volume of product used was different for each study. The timing of outcome measures was variable and could include instantaneous investigations, evaluations every three weeks, or scheduled 4, 8, 12, and 24 weeks after treatment, or evaluations after three months, six months, and one year.

### 3.3. Risk of Bias within Studies

A summary of these evaluations is presented in Figures 2 and 3.

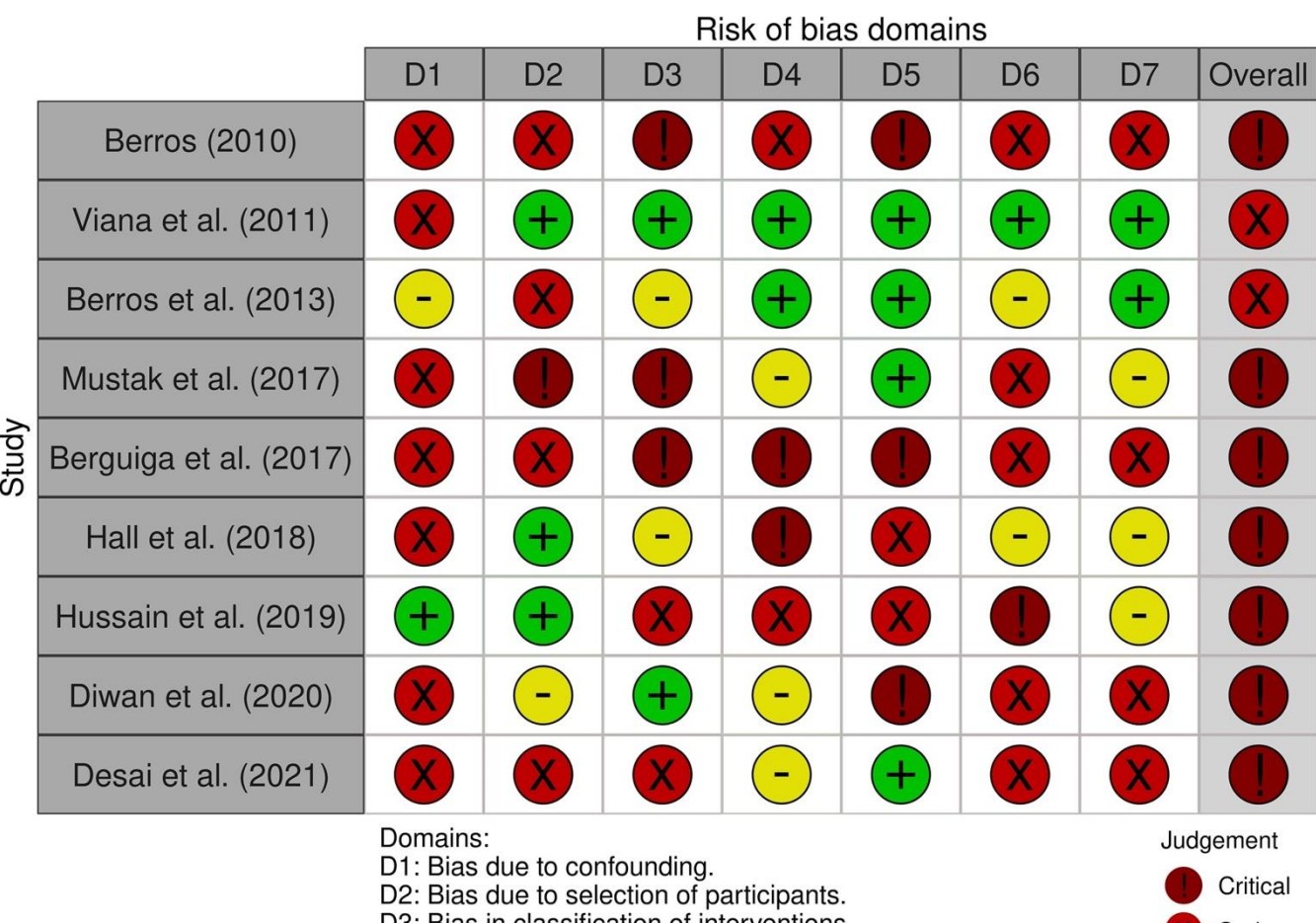

**Figure 2.** ROBINS-I Traffic Light Plot bias assessment.

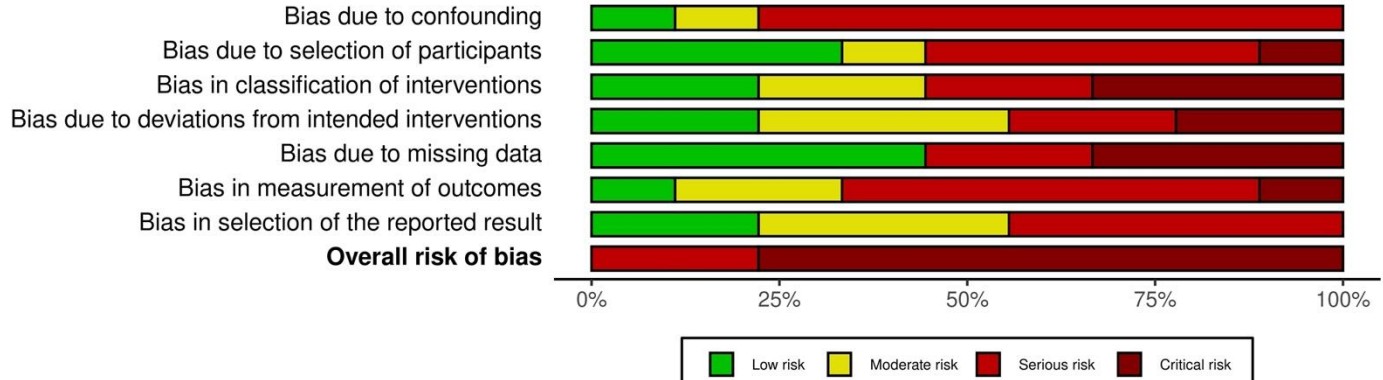

**Figure 3.** ROBINS-I Weighted Summary Plot bias assessment.

### 3.4. Results of Individual Studies

Viana et al. [6], in 2010, treated 25 patients with a serial puncture technique through the 30-gauge needle in the pre-periosteal tissues (total injection volume of filler "Restylane" (Galderma, Fort Worth, TX, USA) for each side was 0.1 to 1.1 mL on the right side and 0.2 to 1.2 mL on the left side) for the tear trough treatment. At the first follow-up (seven

days), they observed bruising in 13 patients (52%), erythema in 10 patients (40%), and local swelling in two patients (8%).

Berros P., in a study conducted from December 2008 to July 2009, treated 26 patients with contour abnormalities in the periorbital region [22]. The hyaluronic acid gel in each case was Restylane. A cannula was used to perform the injection was performed parallel to the periosteum, and, on average, 0.8 to 1.0 mL oh HA was injected. At the follow-up, 7 patients showed hematomas (13%), one patient showed the Tyndall effect (3%), 12 patients showed edema (21%), and 2 patients showed surface irregularities (14%).

Berros et al., in a retrospective study from January 2009 until January 2013, treated 176 patients with tear trough abnormalities [23]. The authors used two methods on two patient collectives. Group A was treated using hyaluronic acid gel (Restylane; Q-Med, Uppsala, Sweden) and a reinforced 25-gauge Pix'l+ micro cannula. The authors developed a modified method for group B that included a combination of cooling of the periorbital area, no local anesthesia, pre-incision displacement of malar fat 10 mm below the orbital border, and postintervention corticoid therapy for 48 h. The quantity of HA injected was 0.6 to 1.0 mL per side, parallel to the periosteum. The complication of edema and swollen reactions were observed in 51.2 percent (21 of 41) of group A. Eleven of 41 patients (26.8 percent) complained about lump or surface irregularities after treatment with protocol A. Alteration of pigmentation was observed in 17.1 percent (seven of 41) in group A. Migration of the injected hyaluronic acid was observed in 16 of 41 patients in group A (39.0 percent). A hyaluronidase injection to minimize complications was necessary for 48.8 percent (20 of 41 patients) in group A. Only group A was evaluated, as group B, in which pre-dressings are used before the injection of HA, did not meet the inclusion criteria of our study.

Berguiga et al., in 2017, treated 151 patients with the use of a semi-cross-linked hyaluronic acid filler "Teosyal® PureSense Redensity (TEOXANE SA, Geneva, Switzerland)" for the tear trough deformity [9]. The procedure was performed using a standard 30-gauge needle in 58% of cases and a cannula in 42% of cases. Injections were administered in the pre-periosteal tissues with a mean volume of 0.48 mL for each side (range, 0.1–1.0 mL). No serious complications were recorded. At the first visit, immediately after the treatment, the 151 patients showed: swelling in 22 patients, bruising in 17 patients, redness in 32 patients, pain in only 1 patient, and blue discoloration in 4 patients. At the 1 month follow-up (visit 2) of 112 patients: swelling occurred in 13 patients, bruising in 12 patients, redness in 7 patients, and blue discoloration in only one patient were recorded.

In a retrospective review of 2017, Mustak et al. evaluated the efficacy and safety of filler injection "Restylane" in 147 patients performed for the rejuvenation of the periorbital tissues, which were followed for at least 5 years since 2014 [24]. All patients underwent injection into the periorbital area using a fanning technique in the suborbicular plane using a 30-gauge needle. The mean number of injections for a patient was 6.88 (±1.72), and the mean total volume injected was 3.19 mL (±0.42). Seventeen patients developed malar edema, three blue-gray dyschromia, and three contour irregularity/orbital ridge. No severe adverse effects were revealed; malar edema, blue-gray dyschromia, and contour irregularity were present in both short-term and long-term follow-up, but malar edema occurred early, often after the first injection.

Hall et al., in a retrospective observational study in 2018, evaluated safety for the treatment of infraorbital hollowing with Juvederm Voluma XC (Allergan Inc., Dublin, Ireland) in 101 patients from February 2016 to March 2017 [25]. They used 27-gauge microcannulas to fill the infraorbital region, and the filler was injected in a supraperiosteal or submuscular plane. The total mean injection volume was 1.0 mL of HA gel. A total of 12 patients (12%) had adverse events related to the injection of Juvéderm Voluma XC. Of those 12 patients, 3 had more than 1 adverse event (25%). Despite the thin skin of the periorbital region, only one patient had the Tyndall effect. Three patients developed diffuse doughy edema of the infraorbital area. Ten patients had bruising and only two had contour irregularities. Most of these adverse events were temporary, with only three patients requiring hyaluronidase to reverse the injection.

Hussian et al., in 2019, in an interventional non-randomized observational study, treated 150 patients with Hyaluronic acid filler gel Juvederm Ultra Plus XC (Allergan Inc., Dublin, Ireland) between January 2017 and February 2018 [26]. They introduced a new procedure for the tear trough treatment based on just three bolus injections called the Tick technique. The volume injected was different according to their grade of depression: 0.3 mL for Hirmand grade 1, 0.4 mL for grade 2, and 0.5 mL for grade 3 using a 31G 6-mm-long needle with perpendicular bolus release at the above periosteal level. Fifteen percent of patients underwent touch-ups for optimal correction with boost injections from 0.1 to 0.2 mL. No serious complications were reveled. Immediately after the injections, 12/150 patients (8.0%) had swelling, and redness was seen in 6/150 patients (4.0%), pain in 3/150 patients (2.0%), and bruising in 3/150 patients (2.0%). After 1 week post-treatment, 7/147 patients had swelling (4.7%), and bruising was recorded in 2/147 patients (1.4%).

Diwan et al., in 2020, injected Teosyal Puresense Redensity 2 in 24 patients for the tear trough treatment. The procedure was performed in the supra-periosteal layer, with injections using exclusively cannulas [27]. In the post-injection period, they observed: moderate swelling in 1 patient, mild swelling in 22 patients, bruising in 1 patient, and pre-syncopal symptoms in 1 patient. At 2 weeks, they observed: mild swelling in six patients, moderate swelling in one patient, bruising in two patients, puffiness in four patients, mild asymmetry in one patient, watery eye in one patient, and overall minimal difference in one patient. At 4 weeks, they observed: swelling in two patients and puffiness in one patient.

Desai et al., in 2021, in a retrospective case note review, treated 165 patients with a hyaluronic acid product, Restylane Vital light (Galderma, Watford, UK), that was injected in the pre-septal region of the tear trough [28]. The manufacturer's supplied syringe was used, which was secured to a 31-gauge 4 mm needle (TSK Mesotherapy Needle). The treatment was performed to a visual endpoint of small subdermal "bubbles" placed at regular intervals of 3–5 mm within the upper 1/3rd of the superomedial tear trough and the pre-septal hollow. A hundred patients noted variable bruising that lasted a median of 3 days (range 2–7 days). All patients recorded having some visible localized eyelid bumps in the treated area, which subsided in all eyelids by 4 days (median), range 2–7 days. Two patients had persistent eyelid swelling. In one of these patients, swelling resolved after 6 weeks to achieve the desired result, whilst the other patient needed the filler dissolved.

Three patients developed Tyndall (blue-grey discoloration) on follow-up (one patient at 3 months, and two patients at 6 months follow-up). No patient experienced infection or blindness.

*3.5. Results of Synthesis*

The extraction of data from the nine evaluated articles allowed us to list a total of 830 patients treated for tear trough deformity with HA injection. Four hundred and fifty-four received injections in the epi-periosteum plane, and 376 were subjected to more superficial injections; 69.2% were treated using needles as a device, and 30.8% using a cannula. Among the 830 participants, no major post-treatment complications were recorded. The complications noted by the authors in relation to the various follow-ups make it difficult to compare and determine percentages. No serious adverse effects were noted. The most frequently observed complications were swelling, bruising, redness, erythema and edema, contour irregularities, and dyschromia, respectively, in 61 cases, 162 cases, 76 cases, 10 cases and 53 cases, 228 cases, and 63 cases. Less frequent adverse effects were: pain in five cases, puffiness in four cases, itching and hollowness in two cases, pre-syncopal symptoms in one case, and watery eye in one case.

The results are summarized in Table 1.

**Table 1.** Results of individual studies.

| Title | Authors | Type of Study | Numbers of Patients | Type of Filler Applied | Volume | Injection Layer | Complications |
|---|---|---|---|---|---|---|---|
| Treatment of the Tear Trough Deformity With Hyaluronic Acid | Giovanni Andrè Pires Viana et al., 2010 | Prospective clinical trial | 25 patients | Restylane (Galderma, Fort Worth, TX, USA) | Total injection volume per side (baseline and touch-ups) was 0.1 to 1.1 mL on the right side and 0.2 to 1.2 mL on the left side. | Pre-periosteal tissues immediately inferior to the orbital rim, with 30-gauge needle. | - Bruising in 13 patients,<br>- erythema in 10 patients<br>- local swelling in 2 patients |
| Tear trough rejuvenation: A safety evaluation of the treatment by a semi-cross-linked hyaluronic acid filler | Berguiga et al., 2017 | Prospective multicenter clinical trial | 151 patients | Teosyal® PureSense Redensity 2 (TEOXANE SA, Geneva, Switzerland) | Mean volume of 0.48 mL for side (range, 0.1–1.0 mL) | - 58% (87 patients) injections with serial puncture in contact with the periosteum, with 30-gauge needle.<br>- 42% (64 patients) retrograde injection technique with cannula deeper than the orbicularis muscle | At the first visit post-treatment on 151 patients:<br>- Swelling in 22 patients (mild 10, moderate 2, missing data 10)<br>- Bruising in 17 patients (mild 9, moderate 2, severe 1, missing data 5)<br>- Redness in 32 patients (mild 12, moderate 3, severe 1, missing data 16)<br>- Pain in 1 patient (mild)<br>- Blued discoloration in 4 patients (moderate 2, missing data 2)<br>- Other (itching, hollow) in 1 patient (1 missing data).<br>At follow-up 1 month (visit 2) on 112 patients<br>- Swelling in 13 patients (mild 4, moderate 2, severe 1, missing data 6)<br>- Bruising in 12 patients (7 mild, moderate 1, missing data 4)<br>- Redness in 7 patients (mild 3, missing data 4)<br>- Blue discoloration in 1 patient (1 missing data)<br>- Other (itching, hollow) in 2 patients (2 missing data) |

**Table 1.** *Cont.*

| Title | Authors | Type of Study | Numbers of Patients | Type of Filler Applied | Volume | Injection Layer | Complications |
|---|---|---|---|---|---|---|---|
| A Prospective Study on Safety, Complications and Satisfaction Analysis for Tear Trough Rejuvenation Using Hyaluronic Acid Dermal Fillers | Diwan et al., 2020 | Prospective study | 24 patients | Teosyal Puresense Redensity 2 (TEOXANE SA, Geneva, Switzerland) | 0.2 to 0.6 mL for side | Supra-periosteal injection using cannula with microdroplet +/− linear threading technique | - Post-injection:<br>- 1/24 moderate swelling<br>- 22/24 mild swelling<br>- 1/24 bruising<br>- 1/24 pre-syncopal symptoms<br>- Average pain score 1.7/10 (number of patients not specified)<br>- At 2 weeks:<br>- 6/24 mild swelling<br>- 1/24 moderate swelling<br>- 2/24 bruising<br>- 4/24 puffiness<br>- 1/24 mild asymmetry<br>- 1/24 watery eye<br>- 1/24 overall minimal difference<br>- At 4 weeks:<br>- 2/24 swelling<br>- 1/24 puffiness |
| Filling the periorbital hollows with hyaluronic acid gel: Long-term review of outcomes and complications | Mustak et al., 2017 | Retrospective case review | 147 patients | Restylane (Galderma, Fort Worth, TX, USA) | Mean 3.19 mL<br>Min 1.2 mL<br>Max 9.7 mL | Fanning technique using a needle 30-gauge in the suborbicularis plane | - 17 patients had malar edema.<br>- 46 blue-gray dyschromia<br>- 45 contour irregularity/orbital ridge |
| The Tick technique: A method to simplify and quantify treatment of the tear trough region | Hussain et al., 2019 | Interventional non-randomized observational study | 150 patients | Juvederm Ultra plus XC (Allergan Inc., Dublin, Ireland) | The volume injected was different according to their grade of depression: 0.3 mL for Hirmand grade 1, 0.4 mL for grade 2, and 0.5 mL for grade 3 | Tick technique based on just three bolus injections at the supraperiosteal level, with 31 gauge needle. | Immediately after injection: −12/150 patients (8.0%) had swelling,<br>- redness in 6/150 patients (4.0%),<br>- pain in 3/150 patients (2.0%),<br>- bruising in 3/150 patients (2.0%).<br>- After 1 week post-treatment:<br>- 7/147 patients had swelling (4.7%)<br>- bruising in 2/147 patients (1.4%) |

**Table 1.** *Cont.*

| Title | Authors | Type of Study | Numbers of Patients | Type of Filler Applied | Volume | Injection Layer | Complications |
|---|---|---|---|---|---|---|---|
| Novel Use of a Volumizing Hyaluronic Acid Filler for Treatment of Infraorbital Hollows | Hall et al., 2018 | Retrospective observational study | 101 patients | Juvederm Voluma XC (Allergan Inc., Dublin, Ireland) | The volume injected was 1.0 mL, 0.5 mL for each side. Touch-up in 18 patients, with 0.9 mL in total (range 0.5–1.0 mL) | Microcannula 27-gauge in the supraperiosteal or submuscular plane | Immediately after injection:<br>- 10/101:Bruising (10%)<br>- 2/101:Contour irregularities(2%)<br>- After 2 weeks:<br>- 3/101: Edema(3%)<br>- 1/101 Tyndal effect (1%)<br>- After 1 month:<br>- 3/101: requiring hyaluronidase (3%) |
| Novel technique of non-surgical rejuvenation of infraorbital dark circles. | Desai et al., 2021 | Retrospective case note review | 165 patients | Restylane Vital light (Galderma, Watford, UK) | Amount of product used range (0.1–0.2 mL for each side) | Needle 31-gauge 4 mm, sub dermal layer, using a serial puncture injection technique | - Bruising 100/165 patients, 60.61%<br>- Tyndall effect 3/165 patients, 1.82%<br>- Eyelid swelling 2/165 patients, 1.21%<br>- Bump 165/165 |
| Periorbital Contour Abnormalities: Hollow EyeRing Management with Hyalurostructure | P. Berros, 2010 | Prospective study | 26 patients | Restylane (Galderma, Fort Worth, TX, USA) | Amount of product used 0.8–1.0 mL. | Microcannula injection 40mm long, parallel to periosteum layer. | - Hematomas 7 (13%)<br>- Pigmentation alteration (dark or blue) 1 (3%)<br>- Edema 12 (21%)<br>- Lump or surface irregularities 2 (14%) |
| Hyalurostructure Treatment: Superior ClinicalOutcome through a New Protocol—A 4-YearComparative Study of Two Methods for TearTrough Treatment | Berros et al., 2013 | Retrospective study | Group A 41 patients | Restylane (Galderma, Fort Worth, TX, USA) | Gentle injection of 0.6 to 1.0 mLof hyaluronic acid per side. | 25-gauge periorbital cannulapenetration until bone contact, followed bypositioning the cannula parallel to the periosteum.Group A: Injection point in rim. | Group A:<br>- Hematomas 11 (26.8)—Edema/swollen reaction 21 (51.2)<br>- Lump/surface irregularities 11 (26.8)<br>- Pigmentation alteration 7 (17.1)<br>- Hyaluronidase injection 20 (48.8) |

## 4. Discussion

The demand for non-surgical procedures to correct blemishes has grown considerably in recent years, and the injection of hyaluronic acid (HA) represents the second most common procedure after the injection of botulinum toxin, with an increase in demand of 60% from 2014 to 2018 [29]. Hyaluronic acid-based treatments offer a valid alternative to some surgical interventions, with immediate results, little or no recovery times, and the possibility to repeat the procedure if needed [30–32]. Treatment options for the tear-trough deformity can be surgical [33], alloplastic implant [34], or autologous fat grafting [35]. An alternative treatment is represented by calcium hydroxyl-apatite (CaHA), which acts as a bio stimulating material, inducing the formation of new collagen with consequent replenishing anointing effect. Unlike hyaluronic acid-based fillers for which we can reverse possible complications through hyaluronidase use, CaHa has no antidote, and its use is therefore recommended to more experienced injectors [36]. Soft tissue augmentation with HA fillers is a common and minimally invasive procedure, although not devoid of possible complications [37,38]. Although the injection of HA products is generally well tolerated, rare serious complications can occur when used to treat tear trough deformities. Accidental retinal artery occlusion by either direct injection or compression is a very rare complication that can occur due to the anatomical complexity of this area [17]. The onset of this complication is related to the presence of numerous branches of the ophthalmic artery in the periocular region, whose direct or indirect involvement during the use of HA fillers can cause blindness [39,40]. However, in 2012, HEXSEL et al. suggested that local adverse events were related to injection techniques (needle or cannula) and not to the different properties of the fillers; various patient factors, the selection of the product to be used, the choice of the injection procedure, and the devices used, are essential to obtain more satisfactory results and reduce the occurrence of adverse effects [41,42].

It is very important to select the appropriate filler, in relation to the anatomical characteristics of the region to be treated, in order to avoid complications, and it is useful to know the rheological properties of the fillers, their physiology, the dimensions and concentrations of the particles and the properties derived from the level of cross-linking of HA. In the treatment of the peri-ocular region, the use of a filler with high $G'$ and low affinity for water such as Restylane (Galderma) reduces the incidence of side effects [43].

In the present study, 830 patients who underwent injections of hyaluronic acid-based fillers for tear trough deformity corrections were analyzed. Of these, 404 patients (48.7%) received hyaluronic acid (HA) gel filler Restylane, 175 patients (21.1%) received Teosyal Puresense Redensity 2, 150 patients (18.1%) received Juvederm Ultra plus XC, and Juvederm Voluma was used in 101 patients (12.1%). In the majority of cases, doses from 0.1 to 1.2 mL per side were administered, precisely in 683 patients (82.2%), while in 147 patients, the dose is higher with an average of 3.1 mL (17.8%). About 69.2% of patients underwent injection through a needle, while cannulas were used in about 30.8%. In 454 patients (54.69%), the filler was placed at the supra-periosteal level, while in 376 patients (45.4%), it was placed in the suborbicular and subcutaneous plane.

The concentrations of hyaluronic acid present in the different products available vary depending on the market; however, high concentrations and the use of large HA particles are related to a higher incidence of soft tissue edema [24] as well as other complications [44]. A higher degree of cross-linking gives the product greater viscosity and allows optimal positioning at the level of the periosteal epi layer, increasing the longevity of the product, reducing surface irregularities as it has a reduced affinity for water; however, it is more commonly correlated with the onset of blue-gray discoloration [9,43–45].

Fast and high-volume injections are more frequently associated with adverse events [10,13,24].

Complications observed in HA tear trough treatment may be related to patient characteristics or to the product or procedure used but are often correlated to multiple factors. Fortunately, despite the possibility of severe but rare complications, most adverse reactions are transient and minor; over 90% of adverse events are related to the injection site,

namely redness, blue-gray discoloration, swelling, contour irregularities, bruising and edema [1,10,46].

The complications noted by the authors during the various follow-ups make it difficult to compare and determine percentages. No serious adverse effects were noted. The most frequently observed complications evidenced in the literature were swelling, bruising, redness, erythema, and edema, blue-gray discoloration, and contour irregularities, respectively, in 61 cases, 162 cases, 76 cases, 10 cases, 53 cases, 63 cases, and 228 cases. Less frequent adverse effects were: pain in five cases, puffiness in four cases, itching and hollow in two cases, pre-syncopal symptoms in one case, watery eye in one case, asymmetry in one case, and minimal difference in one case. A general reduction in adverse reactions was observed in three articles, nevertheless swelling and bruising were still recorded after 2 and 4 weeks up to 1-month post-treatment [9,26,27].

The most frequently encountered complications were edema, swelling, redness, bruising, contour irregularities, and blue-gray discoloration. Edema was mostly seen with the use of cannulas as a delivery system, particularly with cannulae measuring less than 24 gauge in diameter [23,27]. A lower incidence was instead recorded with the use of needles with a diameter greater than 30 gauge [6,9,26]. It is important to note that patients with Hirmand grade 3 laxity, a clinical history of excessive fluid retention, and reduced skin tone generally have a higher degree of post-treatment edema, and therefore a correct pre-procedural evaluation and accurate quantification of the volumes of HA to be injected is necessary [1,39]. In the treatment of tear trough deformities, bruising is one of the three most commonly reported complications following filler injection. Hall et al., in a study of 101 patients, used the cannula injection technique, reporting an incidence of bruising of 10.3% [25]. On the other hand, Desai et al. and Viana et al., who always used a serial needle puncture technique, reported an incidence equal to 60.6% and 52%, respectively [6,28].

Diwan et al. reported bruising in only one patient who was injected with a cannula [27]. Hussain et al. showed a 2% incidence of bruising out of a total of 150 patients when using the needle technique and HA bolus deposition [26]. Berros et al. and Mustak et al. found no cases of bruising in their studies, respectively, with the use of cannula and needle [22–24]. The use of the 22 G cannula in this area is recommended because a small reduced incidence of bruising is correlated with the use of the cannula technique [47]. Despite this, in our study, it emerged that the incidence of this side effect is strictly related to the injection technique and the number of external passages performed, rather than the device used [24,27,47]. It can be seen, in fact, that even using the needle with injection techniques reduces skin trauma (fanning technique, three bolus injection); there is a reduction in the incidence of bruising, with percentages comparable to the use of the cannula [24,26]. Immediate post-procedure swelling is another commonly reported side effect of using fillers in this area. Berguiga and Galatoire reported immediate swelling in 15% of patients in whom the needle technique was chosen [9]. Hussain et al. reported an incidence of 8% out of a total of 150 patients using the needle technique [26]. Berros et al., Mustak et al., and Hall et al. did not report any cases of post-procedure swelling [22–25]. Diwan et al. did not report swelling in any of their patients at 4 weeks following injections with cannula [27]. It is important for this aspect to take into consideration the characteristics of the hyaluronic acid used [48,49]. It is possible to show a higher incidence of swelling when an HA filler with a lower G′ and a lower degree of cross-linking is used (Teosyal, Teoxane) [9,27]. When a product with a high G′ is used, the affinity for water is reduced, and the incidence of swelling is lower (Restylane, Galderma) [22,23].

The needle technique is more associated with the development of swelling, particularly when performed on a more superficial tissue layer [1,10,24,50]. Blue-gray dyschromia, also known as the Tyndall effect, is commonly reported in this area after filler injections [51,52]. The Tyndall effect is a phenomenon that occurs more commonly in patients with thin, poorly pigmented skin, but the exact cause of the discoloration is poorly understood. In order to minimize the occurrence of this complication, injections of HA should not be performed too superficially [50]. The ideal plane of the positioning of small quantities of

filler is below the orbicularis muscle of the eye or at the pre-periosteal level 1. Our studies show that the onset of this effect is not related to the type of filler used or to the delivery system chosen, but it is strictly dependent on the depth of deposition of the HA. When this is deposited at the epi-periosteal level, the incidence is practically nil [6,9,26,27]; instead, when it is placed on a more superficial layer, the occurrence of blue-gray discolorations is higher than 30% [24]. It is also important to specify that the administration of volumes of HA that is too high and the deposition of the filler on superficial planes are to be avoided, as these determine a greater compression of the lymphatic vessels and a greater incidence of the Tyndall effect [10].

Contour irregularities can occur both as early or late manifestations and are observed more frequently above the inferior orbital border [24]. Contour irregularities are most commonly related to injections of excessive volumes, which are often performed too superficially [53,54]. As highlighted in our review, no major surface irregularities were reported in the treated areas when the filler was injected deeply [26,27], while they were seen in a higher percentage of the cases when the filler was injected more superficially [24,25]. Gently massaging the area immediately after the injection can help minimize the presence of surface irregularities; in any case, the gradual dispersion of the filler over time will improve any irregularities. Persistent complications can be resolved by dissolution with hyaluronidase, as in the case of persistent edema or swelling or with the addition of fillers [9,24,25].

It should be pointed out, from what was reported by Berros et al., that the use of a modified protocol used in group B of patients enrolled in their study, which includes pre-injection and post-injection cooling of the periorbital area, incision with the displacement of malar fat 10 mm below the orbital border, some gentle back and forth movements during injection, and oral corticosteroid therapy 48 h after surgery, was found to be more effective and safe, with significantly lower complication rates [23].

## 5. Conclusions

While HA-based fillers cannot eliminate completely tear trough deformities, they can certainly improve them without submitting the patient to surgical interventions. Treating this area with fillers has several advantages: the injection is relatively easy to perform, there is a high degree of patient satisfaction, most complications are often self-limiting, relate to the injection site, and can be easily treated, and in the case of an unsatisfactory effect or persistent complications, the material can be dissolved through the use of hyaluronidase. However, careful pre-treatment patient evaluation is required, and it is advisable to inject HA deeply and in small quantities to reduce the occurrence of complications.

It is important to highlight that, despite having accurate evaluation methods, the scientific studies on the evaluation of medical and cosmetic surgery treatments present a high risk of bias due to serious errors in the selection process of patients, reported results, and confounding factors.

Therefore, it is advisable to increase the accuracy of the results and use appropriate study designs, which allow a real evaluation of the scientific evidence.

**Author Contributions:** Conceptualization, S.D. and S.S.; validation, R.F., R.V. and P.G.; investigation, P.G. and P.B.; data curation, G.L.G.; writing—original draft preparation, S.D.; writing—review and editing, R.V. and G.L.G.; project administration, S.S. All authors have read and agreed to the published version of the manuscript.

**Funding:** This research did not receive any specific grant from funding agencies in the public, commercial, or not-for-profit sectors.

**Institutional Review Board Statement:** Not applicable.

**Informed Consent Statement:** Not applicable.

**Data Availability Statement:** Data are available upon request from the corresponding author.

**Conflicts of Interest:** The authors declare no conflict of interest.

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
