# Peer review of "Is the Treatment of the Tear Trough Deformity with Hyaluronic Acid Injections a Safe Procedure? A Systematic Review"

_applsci, doi:10.3390/app112311489_

Round 1

Reviewer 1 Report

The study design does not lend itself to adequately answer the primary question.

The results are likely to be biased (selection bias), key confounding factors are not addressed.

Author Response

Dear reviewer, our methodological profile has been extensively modified and revised.
We evaluated the safety of HA treatment of the deformities of the periocular region, assessing the incidence of complications related to the treatment and the best technical measures to reduce them, based on the study of the evidence present in the literature

Reviewer 1:

1) The study design does not lend itself to respond adequately to the head physician request.

1) The study. is aimed at evaluating the safety of the treatment of deformities of the. tear trough, by means of HA injections, and evaluate which are the best methods to avoid and reduce the onset of related side effects.

2) Results are likely to be biased (selection bias), a key confounding factor factors are not taken into account.

2) The. article selection has been revised, an article is. been eliminated (Nanda et al.). and 4 new ones were added (P. Berros, 2010, Berros et al., 2013, Hall et al., 2018

Desai et al., 2021), the search queries have been implemented (tear troungh AND volumization, tear trough AND rejuvenation, tear trough AND non-surgical, infraorbital hollowing AND volumization, infraorbital hollowing AND rejuvenation, infraorbital hollowing AND non-surgical) .

Best regards.

Reviewer 2 Report

D’Amato et al presented a manuscript titled „ Is the treatment of the tear trough deformity with hyaluronic acid injections a safe procedure? A Systematic Review”. Even though the topic is interesting the manuscript has many flaws and ambiguities that need to be addressed.

  1. You stated in the Abstract that Galderma Restylane was the most used HA filler. However, I cite you from the Results section: “A total of 198 patients treated with hyaluronic acid filler injections in the lower lid were evaluated; of these, 151 patients were treated with Teosyal PureSense Redensity and 47 with Galderma Restylane.” So was Galderma or Teosyal the most used HA filler? Please revise this.
  2. Moreover, when you bring up branded names please use the scientific manner at the place of the first mention in the manuscript (example: Branded item (Company name, City, Country)).
  3. I can’t agree that the HA filler effects are long lasting, especially when compared to surgical methods. Revise that in your Abstract conclusion.
  4. Please state the time period during which the study was conducted. You only stated the end date.
  5. You should have added to your search keywords the terms: tear trough AND volumization; tear trough AND rejuvenation; tear trough AND non-surgical. Furthermore, you should have used infraorbital hollowing as another term for tear trough since it is called that way in a lot of studies.
  6. You stated that one of the exclusion criteria was not describing the units of HA applied. However, according to your Table 1 the study by Mustak et al. did not specify their volume of HA used. This is a major oversight so please thoroughly check and revise this.
  7. In the 3.1. subsection of the Results you twice stated that 7 articles were included in your review. Furthermore, in this subsection you even explained in detail how did you get to these 7 articles. However, later in the manuscript you only mention 6 articles, both in text as well as in the figures and tables. This needs to be thoroughly revised.
  8. Moreover, the numbers in your 3.1. subsection don’t match the numbers of Figure 1. Additionally, the numbers in text of 3.1. subsection don’t add up to 10836. This is another major oversight.
  9. I do not understand your 3.2. subsection. You state that a total of 198 patients were evaluated, however that does not add up from the 6 studies you included. Moreover, in 3.5. subsection you mention another sample number.
  10. In 3.5 subsection you stated that needles were used in 81% of the included patients while cannula was used in 19%. I have to cite your Abstract: “The injection technique was performed through the use of needles or more frequently with cannula.” How can 19% mean “more frequently”? Please revise this.
  11. Your methodology seems very flawed and it is a major weakness of your work. You need to thoroughly revise and restructure your manuscript. Moreover, due to the previously mentioned oversights I advise you to review all of your data.

Author Response

Dear reviewer, our methodological profile has been extensively modified and revised.

The requested changes have all been made, and the search profile has been implemented.
Technical errors have all been reviewed and corrected.

Reviewer 2:

D'Amato et al presented a manuscript entitled "The treatment of tear deformity with hyaluronic acid injections a safe procedure?

A systematic review ". Although the topic is interesting the manuscript has many flaws and ambiguities that need to be addressed.

1) You stated in the abstract that Galderma Restylane was the HA. most used filler. However, I quote you from the Results section: “A total of 198 patients treated with hyaluronic acid filler injections in the lower eyelid have been evaluated; of these, 151 patients were treated with Teosyal® PureSense Redensity and 47 with Galderma Restylane. "So was Galderma or Teosyal the most used HA filler? Please review this.

Also, when talking about brand names, use the scientific font manner in the place of the first mention in the manuscript (example: Branded item (company name, city, country)).

1) The data has been revised and corrected and the most used filler is [404 with Galderma Restylane]

2) I can't agree that the filling effects of HA are long lasting, mostly compared to surgical methods. Review it in your abstract conclusion.

2) This aspect has been revised and modified: Although the duration of treatment of tear trough deformities with HA fillers is not comparable to surgical treatment, this is minimally invasive, safe procedure, quick to perform and with a high degree of patient satisfaction.

3) Please indicate the time period during which the study was conducted. You only indicated the end date.

3) The period has been correctly specified (January 1st 1957 to 2021)

4) You should have added the terms: ripping through AND to the search keywords volumization; lacrimation And rejuvenation; tearing AND non-surgical. Also, you should have used infraorbital voiding as another term for tearing since it is called that way in a lot of Education.

4) This aspect has been revised and implemented (tear troungh AND volumization, tear trough AND rejuvenation, tear trough AND non-surgical, infraorbital hollowing AND volumization, infraorbital hollowing AND rejuvenation, infraorbital hollowing AND non-surgical.)

5) You claimed that one of the exclusion criteria did not describe the HA units applied. However, according to your Table 1, the study of Mustak et al. did not specify their volume of HA used. This is an important one supervision so please check and review this carefully.

5) This aspect has been carefully revised and missing data has been entered (Mean 3.19 ml  Min 1.2 ml  Max 9.7 ml)

6) In 3.1. Results subsection you have stated twice that 7 articles have been included in your review. Also, in this subsection you can also explained in detail how you came to these 7 articles. However, later on in the manuscript you cite only 6 articles, both in the text and in the in the figures and tables. This needs to be completely revised.

6) This aspect was an oversight, with the new queries the inserted studies were modified (A total of 9 studies were selected as eligible at the end. (Figure 1).)

7)Inoltre, i numeri nel tuo 3.1. la sottosezione non corrisponde ai numeri di Figura 1. Inoltre, i numeri nel testo di 3.1. sottosezione non aggiungere fino a 10836. Questa è un'altra grande svista.

 7)Tali dati sono stati tutti rivisti e modificati:The PubMed search strategy identified 1684 articles for “filler AND complications”, 7602 articles for “HA AND complications”, 138 articles for “tear trough AND complica-tion” and 112 articles for “tear trough AND filler, 89 articles for tear "trough AND hy-aluronic acid", 26 articles for "tear trough AND HA", 44 articles for "tear trough AND (swelling OR bruising OR dyschromia), 13 articles for “tear trough AND volumization”, 4 articles for “tear trough AND non-surgical”. Clinical trials and randomized clinical trials were selected. The Cochrane Library search strategy identified 89 articles using “filler AND complications” as keywords, 1199 when “HA AND complications” was searched, 5 articles for “tear trough and complication” and 7 articles for “tear trough and filler”, 4 articles for “tear trough  AND rejuvenation”,4 articles for “tear trough AND volumi-zation”, 4 articles for “infraorbital hollowing AND volumization”. Trials were filtred. The Ovid search reported no results. The total amount of articles included in the review was 11017. Trials were filtered for each database, case reports, reviews, were excluded,4238 articles were excluded from the research because they were duplicated and 5338 were excluded for others reasons. 1441 records were screened and 507 were excluded because they are out of topic.  934 article were sought for retrieval and of these; 905 articles were screened by title  and 17 were screened for abstract.  5 of them was excluded because not corresponding to inclusion criteria. 7 articles were considered eligible to be included in the review and among the references of these, 2 studies were evaluated and defined as eli-gible for the study. A total of 9 studies were selected as eligible at the end. (Figure 1).

8)Non capisco il tuo 3.2. sottosezione. Dichiari che un totale di 198   i pazienti sono stati valutati, tuttavia ciò non torna dai 6 studi hai incluso.

8)A total of 830 patients treated with hyaluronic acid filler injections in the lower lid were evaluated; of these, 404 with Galderma Restylane ,175 patients were treated with Teosyal PureSense Redensity, 150 patients were treated with Juvederm Ultra Plus XC and 101 patients were treated with Juvederm Voluma. In all studies, the outcome investigated was the enhancement of the tear trough in terms of volume, skin tone, or patients sat-isfaction when treated with hyaluronic acid filler. In the all studies injection layers were defined with precision, the mean volume of product used was different for each study. The timing of outcome measures was variable and could include instantaneous investigations, evaluations every three weeks, or scheduled 4, 8, 12, and 24 weeks after treatment or evaluations after three months, six months, and one year.

9) Furthermore, in 3.5. subsection cite another sample number.

In subsection 3.5 you stated that needles were used in 81% of including patients while the cannula was used in 19%. I have to mention yours Abstract: “The injection technique was performed using needles or more frequently with cannula. "How can 19% mean“ more? frequently "? Please review this.

9) This too has been revised and modified. Section 3.5: 7) Also, the numbers in your 3.1. the subsection does not match the numbers of Figure 1. Furthermore, the numbers in the text of 3.1. subsection do not add up to 10836. This is another big oversight.

 7) These data have all been revised and modified: The PubMed search strategy identified 1684 articles for "filler AND complications", 7602 articles for "HA AND complications", 138 articles for "tear trough AND complication" and 112 articles for " tear trough AND filler, 89 articles for tear "trough AND hy-aluronic acid", 26 articles for "tear trough AND HA", 44 articles for "tear trough AND (swelling OR bruising OR dyschromia), 13 articles for“ tear trough AND volumization ", 4 articles for" tear trough AND non-surgical ". Clinical trials and randomized clinical trials were selected. The Cochrane Library search strategy identified 89 articles using" filler AND complications "as keywords, 1199 when" HA AND complications "was searched , 5 articles for “tear trough and complication” and 7 articles for “tear trough and filler”, 4 articles for “tear trough AND rejuvenation”, 4 articles for “tear trough AND volumes-zation”, 4 articles for “infraorbital hollowing AND volumization. ”Trials were fi ltred. The Ovid search reported no results. The total amount of articles included in the review was 11017. Trials were filtered for each database, case reports, reviews, were excluded, 4238 articles were excluded from the research because they were duplicated and 5338 were excluded for others reasons. 1441 records were screened and 507 were excluded because they are out of topic. 934 article were sought for retrieval and of these; 905 articles were screened by title and 17 were screened for abstract. 5 of them was excluded because not corresponding to inclusion criteria. 7 articles were considered eligible to be included in the review and among the references of these, 2 studies were evaluated and defined as eli-gible for the study. A total of 9 studies were selected as eligible at the end. (Figure 1).

8) I don't understand your 3.2. subsection. You declare that a total of 198 patients were evaluated, however this does not come back from the 6 studies you included.

8) A total of 830 patients treated with hyaluronic acid filler injections in the lower lid were evaluated; of these, 404 with Galderma Restylane, 175 patients were treated with Teosyal PureSense Redensity, 150 patients were treated with Juvederm Ultra Plus XC and 101 patients were treated with Juvederm Voluma. In all studies, the outcome investigated was the enhancement of the tear trough in terms of volume, skin tone, or patients sat-isfaction when treated with hyaluronic acid filler. In the all studies injection layers were defined with precision, the mean volume of product used was different for each study. The timing of outcome measures was variable and could include instantaneous investigations, evaluations every three weeks, or scheduled 4, 8, 12, and 24 weeks after treatment or evaluations after three months, six months, and one year.

Abstract: The injection technique was performed through the use of cannula or more frequently with needle, through the execution of boluses or retrograde release

Best regards

Reviewer 3 Report

While the present article's idea is worthy, the systematic review design is deeply flawed, and the quality of the "Discussion" and "Conclusions" sections lacks depth. The authors should at least rethink the inclusion/exclusion criteria (with care towards the types of the included studies), expand the discussions and draw less subjective conclusions.

Please rephrase the following:

  1. "In fact, it is known that HA injected subcutaneously is absorbed within a year". This is not a known fact; it is only a belief based mainly on data provided by the manufacturers. New studies suggest that the true remanence of hyaluronic acid fillers is actually up to a few years. 
  2. "Fortunately, vascular obstruction is a rare event and can be avoided thanks to a careful knowledge of the anatomical area to be treated and the so-called danger zones." Again, this sounds more like the authors' personal opinion rather than a scientific fact. Unfortunately, vascular obstruction rates are far more frequent than previously reported, and the tear-trough area is one of the most incriminated for this devastating complication (and for blindness as well).

There are some other inaccuracies that need correction.

  1. In the Results section, subsection 3.2 - "Study characteristics", the authors state that "A total of 198 patients treated with hyaluronic acid filler injections in the lower lid 151 were evaluated; of these, 151 patients were treated with Teosyal PureSense Redensity and 47 with Galderma Restylane." We find an obvious contradiction on the number of patients in subsection 3.5 - "Results of synthesis", where we are told that "The extraction of data from the 6 evaluated articles allowed us to list a total of 557 patients treated for tear trough deformity with HA injection." We also find a study, "Hussian et al., in 2019, in an interventional non-randomized observational study 192 treated 150 patients with Hyaluronic acid filler gel Juvederm Ultra plus XC between January 2017 and February 2018.", using a different kind of gel than the ones named in the beginning.
  2. The authors list among the exclusion criteria "articles where numbers of patients, units of hyaluronic acid applied or type of product used were not described." But then, when the included studies are detailed, the authors discuss the "retrospective study of Nanda et al., in 2021, 60 patients were treated with cross-212 linked HA for the correction of periorbital melanosis induced by tear trough deformity[23]. The cross-linked HA filler with low G prime (1 ml for side) was injected in the supra-periosteal area using a needle in 42 patients while a cannula was used in the rest of 215." This study used an unspecified hyaluronic acid filler.

Author Response

Dear reviewer, the requested changes have all been made, and the search profile has been implemented.
Technical errors have all been reviewed and corrected.
More in-depth research was performed on the first 2 points criticized in our manuscript.
The first point has been revised and modified.
The second point was re-evaluated and the evidence found supported our conclusions (added citations of systematic reviews highlighting the low incidence of vascular complications associated with HA treatment, deformities of the peri-ocular region)

Reviewer 3:

Please reword the following:

1) "In fact, it is known that the HA injected subcutaneously is absorbed inside one year". This is not a known fact; it's just a belief based primarily on data provided by manufacturers. New studies suggest that the true the remainder of the hyaluronic acid fillers is actually up to a few years.

1) This aspect has been revised and modified: In fact, it is known that HA injected subcutaneously is absorbed within a 1-3 years and this is closely related to the treated area [11]

2) "Fortunately, vascular obstruction is a rare event and can be avoided thanks to a careful knowledge of the anatomical area to be treated e the so-called dangerous areas. "Again, this sounds more like that of the authors personal opinion rather than a scientific fact. Unfortunately, vascular rates of obstruction are much more frequent than previously reported e the tear-off area is one of the most indicted for this devastating complication (and also for blindness).

2) This aspect has been re-checked and scientifically highlighted with a systematic review evaluation on this topic: Of particular importance, although very rare, are the vascular complications resulting from the intravascular injection of HA [17,18].

3) In the Results section, subsection 3.2 - "Study characteristics", the authors state that "A total of 198 patients treated with hyaluronic acid injections of fillers in the lower eyelid were evaluated 151; of these, 151 patients were treated with Teosyal PureSense Redensity and 47 with Galderma Restylane. "We find an obvious contradiction on the number of patients in subsection 3.5 - "Summary results", where we are told that "The extraction of data from the 6 evaluated articles allowed us to list a total of 557 patients treated for tear deformity with HA injection.

3) These aspects have also been reviewed and implemented: A total of 830 patients treated with hyaluronic acid filler injections in the lower lid were evaluated; of these, 404 with Galderma Restylane, 175 patients were treated with Teosyal PureSense Redensity, 150 patients were treated with Juvederm Ultra Plus XC and 101 patients were treated with Juvederm Voluma. In all studies, the outcome investigated was the enhancement of the tear trough in terms of volume, skin tone, or patients sat-isfaction when treated with hyaluronic acid filler. In the all studies injection layers were defined with precision, the mean volume of product used was different for each study. The timing of outcome measures was variable and could include instantaneous investigations, evaluations every three weeks, or scheduled 4, 8, 12, and 24 weeks after treatment or evaluations after three months, six months, and one year.

4) "We also find a study", Hussian et al., In 2019, in an non-randomized interventional observational study 192 treated 150 patients with Juvederm Ultra plus XC hyaluronic acid filler gel between January 2017 and February 2018. ", using a type of gel other than those mentioned at the beginning.

4) This aspect has been specified section 3.2: 150 patients were treated with Juvederm Ultra Plus XC.

5) The authors list among the exclusion criteria "articles in which numbers of patients, units of hyaluronic acid applied or type of product used were not described. "But then, when the included studies are detailed, the the authors discuss the "retrospective study by Nanda et al., in 2021, 60 patients were treated with HA cross-212 for correction of periorbital melanosis induced by tear deformity [23]. The Low G prime crosslinked HA filler (1ml per side) was injected into the supraperiosteal area using a needle in 42 patients while a cannula it was used in the remainder of 215. "This study used an unspecified hyaluronic acid acid filler.

5) In the absence of the specific requests, this study has been eliminated.

Best regards

Round 2

Reviewer 2 Report

Dear Authors,

You have answered all of my comments and revised the manuscript accordingly. 

Best regards.

Reviewer 3 Report

Congratulations to the authors for the additions they brought to the article.